# Randomised controlled trial to investigate the effectiveness of local oestrogen treatment in postmenopausal women undergoing pelvic organ prolapse surgery (LOTUS): a pilot study to assess feasibility of a large multicentre trial

Tina Sara Verghese ,[1] Lee Middleton,[2] Versha Cheed ,[3] Lisa Leighton,[3] Jane Daniels,[4] Pallavi Manish Latthe,[5] On behalf of the LOTUS trial collaborative group

For numbered affiliations see end of article.

**Correspondence to**
Dr Tina Sara Verghese;
t.s.verghese@bham.ac.uk

## ABSTRACT

**Objective** To evaluate the feasibility of a multicentre randomised controlled trial (RCT) comparing oestrogen treatment with no oestrogen supplementation in women undergoing pelvic organ prolapse (POP) surgery.

**Design and setting** A randomised, parallel, open, external pilot trial involving six UK urogynaecology centres (July 2015–August 2016).

**Participants** Postmenopausal women with POP opting for surgery, unless involving mesh or for recurrent POP in same compartment.

**Intervention** Women were randomised (1:1) to preoperative and postoperative oestrogen or no treatment. Oestrogen treatment (oestradiol hemihydrate 10 μg vaginal pessaries) commenced 6 weeks prior to surgery (once daily for 2 weeks, twice weekly for 4 weeks) and twice weekly for 26 weeks from 6 weeks postsurgery.

**Outcome measures** The main outcomes were assessment of eligibility and recruitment rates along with compliance and data completion. To obtain estimates for important aspects of the protocol to allow development of a definitive trial.

**Results** 325 women seeking POP surgery were screened over 13 months and 157 (48%) were eligible. Of these, 100 (64%) were randomised, 50 to oestrogen and 50 to no oestrogen treatment, with 89 (44/45 respectively) ultimately having surgery. Of these, 89% (79/89) returned complete questionnaires at 6 months and 78% (32/41) reported good compliance with oestrogen. No serious adverse events were attributable to oestrogen use.

**Conclusions** A large multicentre RCT of oestrogen versus no treatment is feasible, as it is possible to randomise and follow up participants with high fidelity. Four predefined feasibility criteria were met. Compliance with treatment regimens is not a barrier. A larger trial is required to definitively address the role of perioperative oestrogen supplementation.

**Trial registration number** ISRCTN46661996.

### Strengths and limitations of this study

► This randomised, external, pilot trial had preplanned feasibility thresholds to judge to assess whether a large trial is feasible.

► The data on number of women at each stage of the trial (screening, randomisation, using the intervention and completing outcome assessments at two follow-up time-points) were collected to estimate recruitment, retention and compliance rates.

► There was the potential for postrandomisation exclusions of participants who were ultimately considered unfit for surgery.

► Blinding of the oestrogen pessary was not possible, so reported outcomes could be subject to responder bias.

► The trial was not designed to be large or long enough to provide convincing evidence for use of preoperative and postoperative oestrogen treatment.

## INTRODUCTION

Pelvic organ prolapse (POP) is the descent of one or more of the anterior vaginal wall, posterior vaginal wall, the uterus (cervix) or the apex of the vagina (after hysterectomy).[1] This definition is based on anatomical change. However, the diagnosis of prolapse should start with symptoms and relate these symptoms to the descent of pelvic organs.[1] Women with symptomatic POP present with a sense of vaginal bulge, in association with a variety of urinary, bowel and sexual symptoms. In postmenopausal women lack of oestrogen is hypothesised to weaken the pelvic floor muscles.[2] While POP surgery is

effective, with symptom improvement in 70%–88% of women, the risk of recurrent prolapse is high, estimated at 10%–30%.[1] Efforts to reduce perioperative complications of POP surgery and enhancement of long-term cure are key priorities for the future management of a rapidly ageing population. In postmenopausal women with vaginal atrophy, surgical dissection may be difficult due to thinning of the vaginal walls. In addition, they may be at a higher risk of surgical wound infection, secondary to changes in the vaginal flora. Oestrogen treatment is often used, on its own or in conjunction with various treatment modalities, for vaginal atrophy.[3] However, despite frequent use, there are no robust data on the benefits of preoperative and postoperative oestrogen treatment in postmenopausal women undergoing POP surgery.

A Cochrane review by Ismail *et al*, published in 2010 did not find any clear evidence to suggest whether oestrogens for the prevention and treatment of POP and concluded an adequately powered randomised controlled trial (RCT) with long-term follow-up is needed to identify the benefits or risks associated with oestrogen supplementation, including before and after POP surgery.[3]

There is a plausible argument for using low-dose oestrogen to improve the vaginal environment and reduce infections at the time of prolapse surgery, but there is little evidence to support its effect on the quality of surgical repair, prolapse cure rates or recurrence. Furthermore, there is no information about any effect in terms of prolapse-related symptoms, overall quality of life (QoL) and outcomes important to women. Finally, the duration of oestrogen treatment, and cost-effectiveness compared with current practice are unknown.

There were over 21 000 pelvic floor repair surgeries performed in England in 2014 on women with an average age of 61 years and incurring a hospital stay of 2 days (HES data, Q08.9, P23.1). These data mask the impact of surgical complications on resource use and QoL, which could be mitigated if perioperative oestrogen was found to be clinically and cost-effective. Moreover, if this use reduces failure and recurrence rates, this would save cost.

Feasibility studies serve to address the question 'can this study be done', by assessing uncertainty in key trial parameters, while pilot studies are a small scale version of the definitive study design to test whether all components can work together. There is often overlap between the objectives of any preparatory studies, as the ultimate aim is to maximise the success of a definitive study by identifying problems with recruitment and data collection processes.[4] The aim of the feasibility study was to find out if an appropriately powered RCT can be realistically undertaken. The overall aim of this research is to determine whether preoperative and postoperative local oestrogen treatment is more effective in improving prolapse-related patient-reported outcomes and reducing recurrence of prolapse symptoms when compared with no treatment. To answer this question, a large, multicentre RCT is required. We report this study as per the Consolidated Standards of Reporting Trials (CONSORT) checklist for reporting a pilot trial.[5]

## METHODS

We conducted a randomised parallel group open external pilot trial of preoperative and postoperative oestrogen treatment versus no treatment in women undergoing POP surgery. An external pilot is a rehearsal of the definitive study where the outcome data are not included as part of the main trial outcome data set.

### Population

Eligible participants were postmenopausal women who were having POP surgery without use of vaginal mesh. Participants were excluded if they had any of the following: hormone replacement therapy in the last 12 months; previous breast/uterine malignancy or other hormone-dependent neoplasms; genital bleeding of unknown origin; previous thromboembolic episodes in relation to oestrogen; known allergy to oestrogens; two or more episodes of culture positive urinary tract infection (UTI) in the last 6 months; voiding dysfunction; previous POP surgery involving mesh or previous POP surgery in the same compartment.

### Study conduct and randomisation

All patients thought to fulfil the eligibility criteria were approached with information regarding the trial. Participants were identified in urogynaecology clinics, provided with a patient information sheet and given the opportunity to consider participation.

Once eligibility was confirmed, the trial team, which comprised urogynaecologists and research nurses from district general hospitals as well as university hospitals across the UK. They conducted face-to-face sessions with the participants in order to obtain written informed consent and baseline data. Women who declined trial participation remained eligible to participate in the qualitative study. Randomisation was performed using a web-based central randomisation system (via Birmingham Clinical Trials Unit) to allocate patients to either oestrogen or no treatment in a 1:1 ratio. Minimisation was used to achieve balance between age (<65 or ≥65 years), parity (≤2 or >2 vaginal births), maximum stage of prolapse (I, II or III/IV) and whether a concomitant continence surgery was planned. An oversight committee was formed to provide independent guidance to the Trial Management Committee and to review accruing safety information during the period of recruitment.

### Interventions

Those allocated oestrogen (oestradiol hemihydrate 10 µg vaginal pessaries; Vagifem, Novo Nordisk) were instructed to use the oestrogen pessaries 6 weeks prior to surgery (once daily for 2 weeks and twice weekly for 4 weeks) up to the night before surgery. The treatment was restarted 6 weeks postoperatively, administering twice weekly for

20 weeks.[6] Patient group discussion expressed that they felt they would be confident to insert an intravaginal pessary 6 weeks after surgery and not earlier as they were worried about disturbing the healing process.

Women were encouraged to insert the pessaries into the vagina at the same time of day. However, if a dose was missed patients were advised it should be administered as soon as possible thereafter, provided the next dose was not due. Participants allocated no treatment received the usual care of the randomising centre. The surgical approach to POP repair was at the discretion of the urogynaecological surgeon. It was not possible to blind clinicians or participants due to the nature of the interventions.

### Feasibility outcomes
#### Primary outcome measure of the feasibility study
To obtain estimates for important aspects of the protocol to allow development of a definitive trial.

#### Secondary outcome measure of the feasibility study
1. Assessment of effectiveness of patient identification and screening processes.
2. Assessment of the effectiveness of the randomisation process of patients.
3. Evaluation of robustness of data collection processes.
4. The proportion of patients followed up at 6 months.
5. Derivation of the preliminary data from clinical outcome measures (eg, Pelvic Floor Distress Inventory Short Form 20 (PFDI-SF20)) to inform the sample size calculation for the substantive study.

### Clinical outcomes
Baseline data including clinical history and symptom-related questionnaires were recorded at the presurgery clinic appointment. Intraoperative, postoperative (6 weeks) and 6 months postsurgery data including objective assessment of prolapse (POP quantification system (POP-Q)), complications (including urinary tract and vaginal infections) and repeat surgery information were collected by the randomising and/or operating clinician at scheduled appointments. The patient-reported questionnaires were: the PFDI-SF20,[7] which comprises three subdomains (Pelvic Organ Prolapse Distress Inventory (POPDI-6), Colorectal-Anal Distress Inventory-8, Urinary Distress Inventory; the Pelvic Floor Impact Questionnaire (PFIQ-7), which also has three subdomains (Pelvic Organ Prolapse Impact-7, Colorectal-Anal Impact-7, Urinary Impact-7); the Pelvic Organ Prolapse/Urinary Incontinence Sexual Questionnaire[8] and Patient Global Impression of Improvement (PGI-I).[9] Questionnaires were collected at baseline then at 6 (at the scheduled appointment) and 12 months, by postal questionnaire. Serious adverse events, reported by clinical staff, were collected throughout the duration of the study.

### Sample size
We planned to randomise 100 women; this number would allow us to measure recruitment and compliance rates with 95% CI between 10% and 20%. It would also be enough to estimate the SD of POPDI-6 with reasonable confidence for future planning of a larger trial (95% CI for SD would be 7 points, assuming the SD is around 20).

### Statistical analysis
Feasibility outcomes were considered with simple summary statistics, with uncertainty estimates provided by 95% CIs. Clinical and patient-reported outcome measures (PROMs) were analysed with point estimates (relative risk or mean difference) and 95% CIs, adjusting for the minimisation variables. This was performed using SAS software, V.9.4 (SAS Institute).

They were not subject to hypothesis testing as the size of the sample would not allow reliable assessment of the effect of the interventions. Participants were considered in the groups they were randomised to regardless of compliance in these summaries (intention-to-treat). The dataset used in this manuscript is available on request from Birmingham Clinical Trials Unit.

### Pilot outcomes
The following outcomes and targets were set a priori as being indicative that a larger trial would be feasible to conduct. These were: (i) patient eligibility rate (the proportion of screened patients eligible) at least 25%; (ii) patient recruitment rate (the proportion of eligible patients randomised) at least 25%; (iii) compliance rate (the proportion of patients with good compliance to treatment, that is, at least 75% allocated pessaries used): at least 50%; (iv) data completion rate (number of follow-up questionnaires completed at 6 months): at least 75%.

### Patient and public involvement
Throughout the process of the development and the running of the feasibility study, the study team have actively involved participation from patients and public in order to refine the trial and make the study about postmenopausal women suffering from symptoms of POP. The research team conducted a qualitative study to identify the motivations for, and barriers to recruitment and participation in LOTUS. Twenty women were interviewed which provided key insight into trial recruitment barriers. This application benefited from the qualitative study in several ways (eg, approaching patients and the acceptability of the investigational medicinal product).

Focus group interviews were held through which the research team gained insight from women who were given the opportunity to express their views and felt that further research in this area is much needed.

### RESULTS
#### Patients and follow-up
Recruitment took place over a 13-month period between July 2015 and August 2016 in six urogynaecology centres in the UK (Birmingham Women's Hospital, Croydon

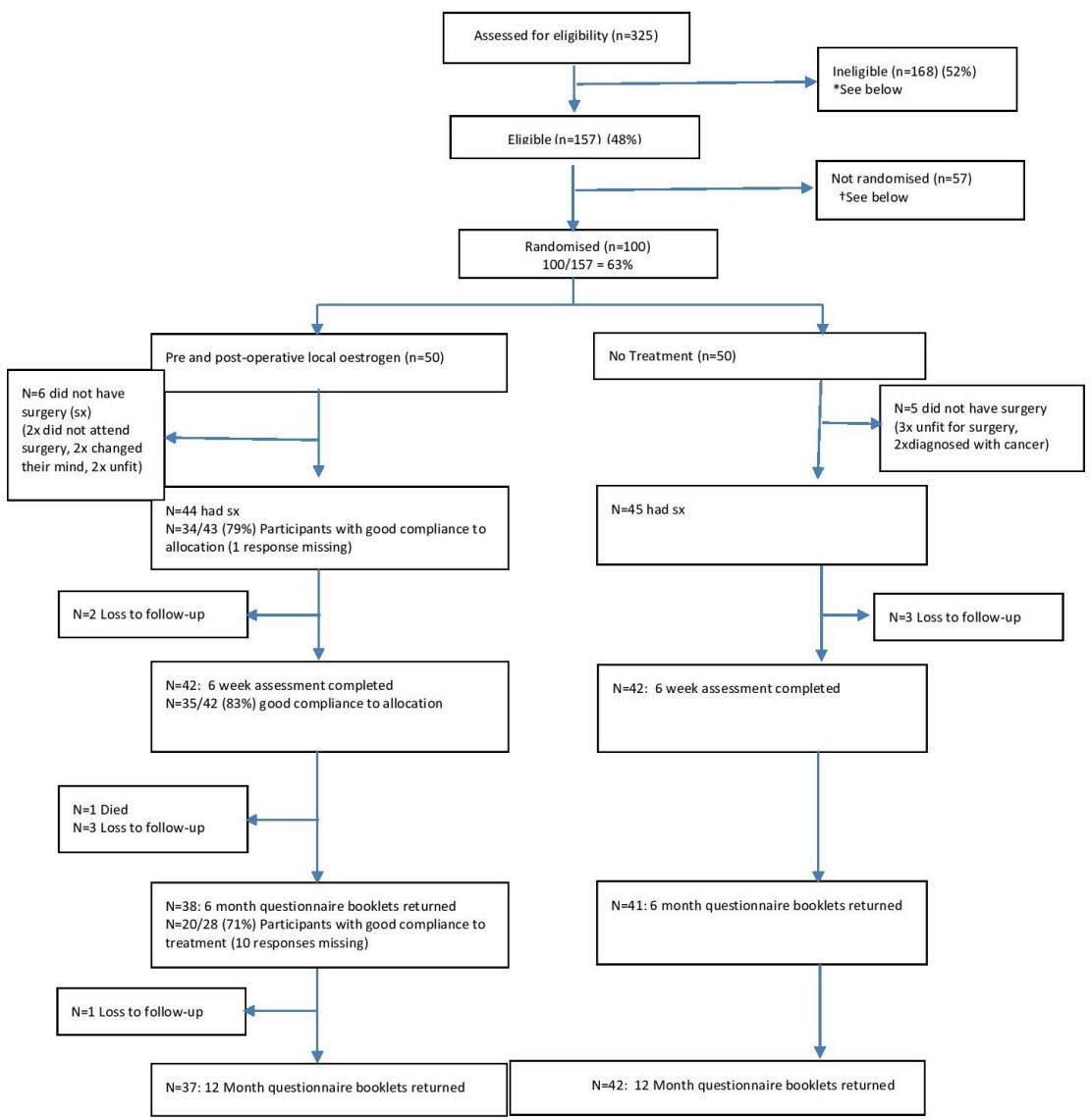

**Figure 1** Flow of participants through the trial. *Reasons for ineligibility N=168 (%): no prolapse surgery required 38 (22), recurrence of prolapse in the same compartment 32 (19), has had previous breast or uterine malignancy 20 (12), undergoing hormone replacement therapy (HRT) treatment 17 (10), unable to understand English 16 (9), genital bleeding of unknown origin 15 (9), Has had previous hormone-dependent neoplasms 13 (8), Previous pelvic organ prolapse (POP) surgery involving mesh 6 (3), Previous thromboembolic episode related to oestrogen therapy 4 (2), two or more culture positive urinary tract infection (UTI) episodes in the last 6 months 1 (<1), other 6 (3).†Reasons for non-randomisation N=57 (%): patient did not attend clinic 19 (33), not enough time to discuss trial and gain informed consent 14 (25), declined consent/not willing to be randomised 9 (16), not willing to wait 6 weeks for surgery 5 (9), unable to gain consent 4 (7), did not want to complete questionnaires, 1 (2).

University Hospital, Basingstoke and North Hampshire Hospital, Walsall Manor Hospital, Royal Stoke University Hospital, James Cook University Hospital, Middlesbrough). Once the target sample size was reached, the study stopped randomising women.

Of the women who presented with prolapse and wanted surgery, 325 consecutive women were screened for eligibility, 157 were found to be eligible (48%, 95% CI 40% to 56%) and 100 (64%, 95% CI 56% to 72%) of those eligible were randomised (figure 1). The minimisation algorithm provide appropriate balance for the minimisation factors; the average age of participants was 66 years with 47% having parity >2 with an average BMI of 28.2. The maximum stage of prolapse was III or IV (increased severity) for 40% of participants (table 1).

Of the 100 randomised women, 89 ultimately had surgery (details given in table 2). Of the other 11, 5 could not have surgery due to health issues, 2 changed their mind about surgery, 2 did not attend their surgical appointment and 2 were diagnosed with cancer. Six-month follow-up questionnaires were completed by 79/100 (79%) of participants. Of those that had surgery this equated to 79/89 (89%). Of those who had surgery, 100% had completed intraoperative data and 84/89 (94%) had postoperative data recorded at 6 weeks.

**Table 1**  Baseline characteristics

| Characteristic | | Oestrogen (n=50) | No treatment (n=50) | Overall (n=100) |
|---|---|---|---|---|
| Age, years* | <65 | 21 (42%) | 21 (42%) | 42 (42%) |
| | ≥65 | 29 (58%) | 29 (58%) | 58 (58%) |
| | Mean (SD) | 65.7 (8.2) | 65.9 (8.4) | 65.8 (8.3) |
| Ethnic group | White | 45 (90%) | 43 (86%) | 88 (88%) |
| | Asian | 2 (4%) | 3 (6%) | 5 (5%) |
| | Black | 3 (6%) | 1 (2%) | 4 (4%) |
| | Mixed | 0 (-) | 2 (4%) | 2 (5%) |
| | Other ethnic group | 0 (-) | 1 (2%) | 1 (1%) |
| BMI (mg/k$^2$) | Mean (SD) | 28.1 (5.1) | 28.2 (5.9) | 28.2 (5.5) |
| | Missing (N) | 2 | 7 | 9 |
| Parity* | ≤2 | 27 (54%) | 26 (52%) | 53 (53%) |
| | >2 | 23 (46%) | 24 (48%) | 47 (47%) |
| Maximum stage of prolapse* | I | 7 (14%) | 7 (14%) | 14 (14%) |
| | II | 23 (46%) | 23 (46%) | 46 (46%) |
| | III/IV | 20 (40%) | 20 (40%) | 40 (40%) |
| Concomitant continence surgery performed* | N (%) | 2 (4%) | 3 (6%) | 5 (5%) |
| Previous operation for prolapse | N (%) | 4 (8%) | 4 (8%) | 8 (8%) |
| Anterior | No. of repairs, median (IQR) | 1 (1–1) | 1 (1–1) | 1 (1–1) |
| Posterior | No. of repairs, median (IQR) | 1 (1–1) | 1 (1–1) | 1 (1–1) |
| Hysterectomy | N (%) | 12 (24%) | 8 (16%) | 20 (20%) |
| Vaginal pessary/ring currently in place | N (%) | 5 (10%) | 8 (16%) | 13 (13%) |
| Physiotherapy treatment for prolapse/ urinary incontinence in last 12 months | N (%) | 8 (16%) | 10 (20%) | 18 (18%) |
| Drug treatment for urinary incontinence | N (%) | 0 (-) | 4 (8%) | 4 (4%) |
| Treatment for overactive bladder | N (%) | 0 (-) | 0 (-) | 0 (-) |

*Minimisation variable.
BMI, body mass index.

## Compliance

Good compliance with oestrogen treatment (denoted as pessaries used at least ≥75% of the expected time) was observed in 79% of participants preoperatively (34/43), 83% at 6 weeks (35/42) and 71% at 6 months (20/28).

## Clinical and patient-completed outcomes

Overall, scores from the PFDI-SF20 (POPDI-6 domain) were low at both 6 and 12 months, averaging 14.3 (SD 16.4) and 15.6 (SD 18.1) out of a maximum of 100, indicating a low level of prolapse-related symptoms at these times. Responses from the PFIQ-7 (POP-IQ-7 domain) were similarly low (table 3). Scores appeared similar in both groups, but with expected high levels of uncertainty given the limited size of sample. The number of participants reporting being improved (very much better or better) on the PGI-I was 92% (73/79) at 6 months and 89% (70/79) at 12 months (table 4). The number of

participants with an objective rating of prolapse failure (from POP-Q) at 6 months was 22/55 (40%) (table 5). Two repeat incontinence surgeries were recorded in the no treatment group over 6 months, with none in the oestrogen group.

## Safety

More UTIs (8/42; 19% vs 4/42; 10%)—resulting in more antibiotics prescriptions (9/42; 21% vs 5/42; 12%) were reported in the 'no treatment' group than in the oestrogen group. Two serious adverse events were recorded in the oestrogen group—one woman with high temperature was admitted for intravenous antibiotics but culture of vaginal and urine samples were negative, and another woman was diagnosed with leukaemia. Neither were thought to be related to treatment. Two incidental hospitalisations were also recorded in the 'no treatment' group (one case of heavy bleeding thought to be unrelated to surgery

| Table 2   Operative details | Oestrogen (n=44) | No treatment (n=45) |
|---|---|---|
| **Aspect of surgery** | | |
| POP surgery performed | | |
| Anterior repair | 35 (80%) | 34 (76%) |
| Posterior repair | 16 (36%) | 19 (42%) |
| Vaginal hysterectomy±bilateral salpingo-oophorectomy | 26 (59%) | 25 (56%) |
| Sacrospinous fixation | 6 (14%) | 6 (13%) |
| Concomitant surgery | | |
| Sling/Tension-free vaginal tape | 1 (2%) | 1 (2%) |
| Botulinum toxin A | 1 (2%) | 1 (2%) |
| Suprapublic catheter | 1 (2%) | 1 (2%) |
| Ease of dissection | | |
| Very easy | 6 (14%) | 5 (11%) |
| Easy | 22 (50%) | 13 (29%) |
| Normal | 14 (%) | 21 (47%) |
| Difficult | 2 (5%) | 5 (11%) |
| Very difficult | 0 (-) | 1 (2%) |
| Blood loss (g) | | |
| N; median (IQR) | n=44; 63 (50–125) | n=44; 54 (50–100) |
| Missing | 0 (-) | 1 (2%) |
| Visceral injury | | |
| Bladder injury | 0 (-) | 0 (-) |
| Urethral injury | 0 (-) | 0 (-) |
| Ureteric injury | 1 (2%) | 0 (-) |
| Bowel injury | 0 (-) | 0 (-) |
| Button hole of vagina | 3 (7%) | 1 (2%) |
| Suture used—fascial plication* | | |
| Polyglactin (Vicryl, Ethicon/Polysorb, Medtronic) | 10 (23%) | 10 (22%) |
| Polydioxanone (PDS-II, Ethicon) | 35 (80%) | 34 (76%) |
| Poligelcarprine (Monocryl, Ethicon) | 0 (-) | 0 (-) |
| Missing | 0 | 1 |
| Suture used—Vagina* | | |
| Polyglactin (Vicryl, Ethicon/Polysorb, Medtronic) | 41 (93%) | 42 (93%) |
| Polydioxanone (PDS-II, Ethicon) | 5 (11%) | 6 (13%) |
| Poligelcarprine (Monocryl, Ethicon) | 3 (7%) | 1 (2%) |
| Vaginal packing used | 23 (52%) | 28 (62%) |
| Cather inserted in theatre | | |
| Yes | 43 (98%) | 42 (93%) |
| Postoperative complications noted before discharge | | |

Continued

| Table 2   Continued | Oestrogen (n=44) | No treatment (n=45) |
|---|---|---|
| **Aspect of surgery** | | |
| Infection | 1 (2%) | 3 (7%) |
| Ureteric injury | 0 (-) | 0 (-) |
| Bladder injury | 0 (-) | 0 (-) |
| Bowel injury | 0 (-) | 0 (-) |
| Vascular injury | 0 (-) | 0 (-) |
| Neurological injury | 0 (-) | 0 (-) |
| Perioperative or postoperative blood transfusion | 0 (-) | 0 (-) |
| Perioperative or postoperative thromboembolism | 0 (-) | 0 (-) |
| Death | 0 (-) | 0 (-) |

*More than one can be selected, hence numbers do not add up to total.
POP, pelvic organ prolapse.

and another with pancreatic cancer). No concerns were expressed by the independent oversight committee who met half way through the recruitment period to review the safety data.

## DISCUSSION

In this open-label randomised study, we sought to examine the feasibility of randomising postmenopausal women undergoing POP surgery to receive vaginal oestrogen. Our four key feasibility indicators were met and we have shown that a large multicentre RCT is feasible. It is possible to randomise and follow-up patients with high fidelity over at least 12 months. Oestrogen treatment in the form of vaginal pessaries was well tolerated and consistently applied by most women. As this was a feasibility study with an open-label design, no inferences can be made about the treatment's therapeutic efficacy. However, the data do suggest that further research is warranted. There was evidence that the majority of women in both trial groups did report improvement in their prolapse symptoms, QoL and other aspects of urinary and sexual function. The effect was sustained for at least the first 6 months after surgery irrespective of oestrogen treatment.

Among women planning to undergo surgical repair for prolapse, three studies have reported on preoperative vaginal oestrogen by various methods, compared with placebo or no treatment, with a total of 111 participants.[10–12] Follow-up varied from 12 weeks to 3 years of postoperative surveillance. No vaginal oestrogen was given following surgery. The overall quality of evidence was poor. Use of vaginal oestrogen improved the vaginal maturation index, a histological measure of the status of the vaginal epithelium, at the time of surgery[13] and

**Table 3**  Results of patient-reported outcomes—PFDI-SF20, PFIQ-7 and PISQ-12

| Outcome measure Timepoint | Oestrogen N; mean (SD) | No treatment N; mean (SD) | Mean difference between groups (95% CI)‡ |
|---|---|---|---|
| PFDI-SF20; POPDI-6 domain (0–100, higher=worse pain)* | | | |
| Baseline | n=48; 43.4 (24.8) | n=49; 46.0 (24.4) | |
| 6 months | n=38; 16.3 (20.3) | n=41; 12.5 (11.6) | 4.1 (–3.9 to 12.1) |
| 12 months | n=37; 16.7 (20.6) | n=41; 14.6 (15.6) | 4.4 (–4.7 to 13.5) |
| PFDI-SF20; CRADI-8 domain (0–100, higher=worse pain)* | | | |
| Baseline | n=48; 26.0 (25.0) | n=48; 24.3 (18.0) | |
| 6 months | n=38; 12.7 (14.4) | n=41; 15.2 (13.2) | –2.4 (–7.6 to 3.4) |
| 12 months | n=37; 12.8 (14.8) | n=42; 15.8 (15.8) | –1.7 (–8.0 to 4.6) |
| PFDI-SF20; UDI-6 domain (0–100, higher=worse pain)* | | | |
| Baseline | n=48; 34.8 (22.6) | n=49; 37.9 (26.7) | |
| 6 months | n=38; 16.4 (17.5) | n=41; 17.2 (18.5) | 0.5 (–8.9 to 7.9) |
| 12 months | n=36; 20.3 (17.5) | n=42; 21.5 (25.3) | 0.7 (–9.1 to 10.4) |
| PFDI-SF20; summary score (0–300, higher=worse pain)* | | | |
| Baseline | n=48; 104.3 (62.8) | n=48; 107.0 (57.4) | |
| 6 months | n=38; 45.4 (43.8) | n=41; 45.0 (37.1) | 0.6 (–18.4 to 17.2) |
| 12 months | n=36; 49.8 (44.1) | n=41; 51.1 (47.7) | 4.8 (–16.2 to 25.8) |
| PFIQ-7; UIQ-7 domain (0–100, higher=worse condition)* | | | |
| Baseline | n=50; 21.0 (25.8) | n=50; 21.5 (24.6) | |
| 6 months | n=38; 9.6 (21.7) | n=41; 5.9 (11.2) | 1.3 (–6.9 to 9.5) |
| 12 months | n=37; 8.9 (16.8) | n=42; 8.0 (13.9) | 1.4 (–6.0 to 8.8) |
| PFIQ-7; CRAIQ-7 domain (0–100, higher=worse condition)* | | | |
| Baseline | n=50; 12.0 (21.0) | n=50; 8.7 (13.5) | |
| 6 months | n=38; 4.9 (13.1) | n=41; 2.4 (6.1) | 1.6 (–2.5 to 5.8) |
| 12 months | n=36; 4.5 (16.3) | n=42; 2.9 (6.1) | 2.2 (–2.8 to 7.1) |
| PFIQ-7; POPIQ-7 domain (0–100, higher=worse condition)* | | | |

Continued

| Outcome measure<br>Timepoint | Oestrogen<br>N; mean (SD) | No treatment<br>N; mean (SD) | Mean difference between<br>groups (95% CI)‡ |
|---|---|---|---|
| **Table 3** Continued | | | |
| Baseline | n=50;<br>17.7 (21.2) | n=50;<br>17.1 (22.1) | |
| 6 months | n=38;<br>5.0 (13.4) | n=41;<br>2.3 (6.1) | 2.5 (−2.5 to 7.5) |
| 12 months | n=36;<br>1.9 (6.8) | n=42;<br>1.7 (5.2) | 0.2 (−2.7 to 3.1) |
| PFIQ-7; summary score (0–300, higher=worse pain)* | | | |
| Baseline | n=50;<br>50.8 (54.8) | n=50;<br>47.3 (51.9) | |
| 6 months | n=38;<br>19.5 (39.8) | n=41;<br>10.7 (19.8) | 6.2 (−8.7 to 21.2) |
| 12 months | n=36;<br>14.3 (29.1) | n=42;<br>12.7 (18.3) | 4.5 (−6.8 to 15.8) |
| | | | |
| PISQ-12 (0–48, higher=better sexual function)† | | | |
| Baseline | n=18;<br>31.2 (6.5) | n=19;<br>33.1 (7.3) | |
| 6 months | n=11;<br>33.1 (4.1) | n=14;<br>36.1 (5.7) | −2.4 (−7.0, 2.2) |
| 12 months | n=11;<br>34.5 (2.7) | n=10;<br>32.8 (5.1) | 3.4 (0.6 to 6.2) |

*<0 indicates less pain/better condition failures with oestrogen.
†>0 indicates better with oestrogen.
‡Differences are adjusted for baseline score and minimisation variables.
CRADI-8, Colorectal-Anal Distress Inventory; CRAIQ-7, Colorectal-Anal Impact; PFDI-SF20, Pelvic Floor Distress Inventory Short Form 20; PFIQ-7, Pelvic Floor Impact Questionnaire; PISQ-12, Pelvic Organ Prolapse/Urinary Incontinence Sexual Questionnaire; POPDI-6, Pelvic Organ Prolapse Distress Inventory; UDI-6, Urinary Distress Inventory.

increased vaginal epithelial thickness.[9] Vaginal oestrogen application for 6 weeks preoperatively increased synthesis of mature collagen and increased thickness of the vaginal wall, suggesting this intervention improves both the substrate for suture placement at the time of repair and maintenance of tissue integrity of the pelvic floor.[10] In our study, we did not monitor the vaginal maturation index but we did note that there were fewer UTIs observed in the group that received vaginal oestrogen preoperatively, although the study was not powered for this outcome. The clinicians reported easier planes of dissection in women in the intervention group.

Mikkelsen *et al* described the 3-year postoperative outcomes after preoperative treatment with vaginal oestradiol or placebo tablets before POP repair.[11] They included a questionnaire of patient satisfaction, but no standardised or validated metric of prolapse symptoms, discontinuation rates and reasons were reported.[10] They suggested that if vaginal oestrogen does improve tissue integrity then therapy may need to be continued post-operatively until the time of complete scar maturation.[12]

There is no clear evidence to state how long when a women would have adequate scar maturation after a prolapse surgery as this could be dependent on various factors of wound healing. In the current study, women in the intervention group continued oestrogen up to 20 weeks postsurgery. The overall improvement in symptoms was comparable at 6 and 12 months postsurgery in both groups. The majority of participants undergoing prolapse surgery in both groups demonstrated improvements in prolapse symptoms (90%) at 12 months, indicating that the benefit of vaginal oestrogen may be marginal, with the majority of improvement in patient symptoms pertaining to surgery. This is hypothetical as there has been no RCT with long-term follow-up on these patients randomised to postoperative local oestrogen versus placebo or no treatment. Karp *et al* found that early administration of vaginal oestrogen after vaginal surgery via an oestradiol-releasing ring is feasible and results in improved markers of tissue quality postoperatively compared with placebo and controls.[14]

**Table 4** Results of patient-reported outcomes—Patient Global Impression of Improvement (PGI-I)

| Timepoint PGI-I response | | Oestrogen | No treatment | Relative risk (95% CI)‡ |
|---|---|---|---|---|
| 6 weeks | | n=42 | n=42 | |
| 1=very much better | N (%) | 22 (52%) | 14 (33%) | |
| 2=much better | N (%) | 16 (38%) | 18 (43%) | |
| 3=a little better | N (%) | 4 (10%) | 8 (19%) | |
| 4=no change | N (%) | 0 (-) | 2 (5%) | |
| 5=a little worse | N (%) | 0 (-) | 0 (-) | |
| 6=much worse | N (%) | 0 (-) | 0 (-) | |
| 7=very much worse | N (%) | 0 (-) | 0 (-) | |
| Better score* | N (%) | 42 (100%) | 40 (95%) | 1.05 (0.98 to 1.12) |
| Worse score† | N (%) | 0 (-) | 2 (5%) | |
| 6 months | | n=38 | n=41 | |
| 1=very much better | N (%) | 13 (34%) | 18 (44%) | |
| 2=much better | N (%) | 16 (42%) | 17 (41%) | |
| 3=a little better | N (%) | 4 (11%) | 5 (12%) | |
| 4=no change | N (%) | 3 (8%) | 0 (-) | |
| 5=a little worse | N (%) | 2 (5%) | 1 (2%) | |
| 6=much worse | N (%) | 0 (-) | 0 (-) | |
| 7=very much worse | N (%) | 0 (-) | 0 (-) | |
| Better score* | N (%) | 33 (87%) | 40 (98%) | 0.89 (0.78 to 1.02) |
| Worse score† | N (%) | 5 (13%) | 1 (2%) | |
| 12 months | | n=37 | n=42 | |
| 1=very much better | N (%) | 14 (38%) | 17 (40%) | |
| 2=much better | N (%) | 15 (41%) | 17 (40%) | |
| 3=a little better | N (%) | 3 (8%) | 4 (10%) | |
| 4=no change | N (%) | 4 (11%) | 2 (5%) | |
| 5=a little worse | N (%) | 1 (3%) | 0 (-) | |
| 6=much worse | N (%) | 0 (-) | 2 (5%) | |
| 7=very much worse | N (%) | 0 (-) | 0 (-) | |
| Better score* | N (%) | 32 (86%) | 38 (90%) | 0.96 (0.81 to 1.12) |
| Worse score† | N (%) | 5 (14%) | 4 (10%) | |

*Better score is derived from scores 1 (very much better) to 3 (a little better).
†Worse score is derived from scores 4 (no change) to 7 (very much worse).
‡>1 indicates better with oestrogen.

Surgical failures requiring repeat POP repairs on an average occur within 2 years after primary surgery.[12] The PROSPECT study reported that 2% (6/395 standard repair) repeat surgeries were required as early as 1 year after primary surgery and 5% (16/348 standard repair) repeat surgery within 2 years.[15] The authors concluded that follow-up of patients for a minimum of 5 years would be required to assess recurrence of symptoms following POP surgery.[15] We need to consider the pros and cons of long-term maintenance treatment with vaginal oestrogen in the intervention group in a definitive study including cost, compliance, plateauing of benefits and regression of changes on stopping the treatment. The potential outcomes measured in long-term studies may include recurrence rates, interval of recurrence of prolapse symptoms from index surgery, reduction in urinary tract symptoms and sustained QoL improvements and cost-effectiveness analysis.

### Strengths

This feasibility study was conducted over a period of 13 months during which the trial achieved its target recruitment of 100 women from six centres in the UK. The study met all the target criteria within the given time frame and showed that a substantial trial could be achieved.

**Table 5** Results of POP-Q assessment

| Anatomical site in vagina Timepoint | Oestrogen N; mean (SD) | No treatment N; mean (SD) | Mean difference between groups (95% CI) |
|---|---|---|---|
| **Aa (anterior wall), cm** | | | |
| Baseline | n=49; 0.1 (1.9) | n=48; −0.3 (1.7) | |
| 6 months* | n=26; −1.7 (2.0) | n=29; −2.6 (2.0) | 0.7 (0.1 to 1.3) |
| **Ba (anterior edge), cm** | | | |
| Baseline | n=49; 0.2 (2.0) | n=48; 0.1 (1.8) | |
| 6 months* | n=25; −2.0 (2.0) | n=29; −2.9 (1.9) | 0.6 (0.0 to 1.2) |
| **C (cervix/cuff), cm** | | | |
| Baseline | n=49; −3.7 (2.5) | n=48; −3.4 (3.4) | |
| 6 months* | n=25; −2.9 (2.8) | n=29; −2.2 (3.6) | −0.2 (−1.3 to 0.9) |
| **Ap (posterior wall), cm** | | | |
| Baseline | n=49; −1.2 (1.5) | n=48; −1.0 (1.7) | |
| 6 months* | n=26; −0.8 (1.7) | n=29; −1.0 (1.7) | 0.5 (−0.2 to 1.1) |
| **Bp (posterior edge), cm** | | | |
| Baseline | n=49; −1.1 (1.6) | n=48; −1.0 (1.8) | |
| 6 months* | n=26; −0.9 (1.8) | n=29; −1.0 (1.8) | 0.4 (−0.3 to 1.1) |
| **D (posterior fornix), cm** | | | |
| Baseline | n=39; −5.3 (1.9) | n=40; −5.1 (2.6) | |
| 6 months* | n=7; −1.3 (1.5) | n=4; −1.8 (1.9) | 0.7 (−6.2 to 7.5) |
| **Total vaginal length, cm** | | | |
| Baseline | n=49; 8.5 (1.1) | n=48; 8.0 (1.6) | |
| 6 months† | n=26; −0.2 (1.3) | n=29; −0.5 (1.7) | 0.3 (0.4 to 0.9) |
| **Objective prolapse (any of Aa, Ba, C, Ap, Bp or D >0) at 6 months** | | | |
| | N (%) | N (%) | Relative risk (95% CI)‡ |
| Failure | 26-Sep | 13/29 | 0.77 (0.40 to 1.50) |

*Change from baseline adjusted for baseline; difference <0 indicates better with oestrogen.
†Adjusted for baseline length; difference <0 indicates better with oestrogen.
‡<1 indicates less prolapse failures with oestrogen.
POP, pelvic organ prolapse; POP-Q, POP quantification system.

We were able to capture patient responses by using validated PROMs such as the PFDI-SF20, PFIQ-7 and PSIQ-12. The feasibility study has been reported in line with the CONSORT reporting guideline for feasibility/pilot studies. The feasibility study has allowed the trial team to identify areas in the study process that require fine-tuning ahead of a larger trial. We are considering long-term follow-up of participants for a minimum of 5 years in order to capture the recurrence rates and relapse of POP symptoms after index surgery. In addition, we are considering offering participants in the intervention group to continue with long-term maintenance therapy with vaginal oestrogen.

### Limitations

Ideally, we would have liked to conduct an RCT comparing vaginal oestrogen with a placebo pessary. All of the symptom-related outcome measures are patient-completed and potentially prone to reporting bias if the participant is aware of the treatment allocation. However, despite extensive discussions with clinical trial suppliers, we were unable to procure a placebo, without it being prohibitively expensive. Oestrogen pessaries are supplied in individually packaged, single-use plastic applicators, which cannot be replicated with a placebo due to the trademark on the packaging. Disassembling the pessary from the applicator to repackage in an unbranded applicator and container would require additional stability testing to confirm the bioavailability of the oestradiol hemihydrate. The manufacturer of Vagifem, Novo Nordisk, declined to provide assistance. Oestradiol is also available as a cream, and a placebo cream could be procured, but the acceptability and adherence to a cream as the route of treatment was not evaluated in this pilot and we should not extrapolate adherence based on pessaries.

A few patients postrandomisation could not undergo surgery as they were deemed to be unfit. To avoid this from happening, the patients could have undergone preoperative assessments to determine surgical and anaesthetic risk and randomised once they were scheduled for surgery. We could potentially have sent telephone, email or text reminders to patient to improve compliance with the postoperative pessary use but it would be expensive and not reflect a real-life situation.

### CONCLUSION

At this stage, we cannot expect that the outcomes of this pilot study to be directly translated into clinical care as the study was not large enough to be able to detect small-to-moderate realistic-sized difference in rates of prolapse-related QoL, nor was the scope wide enough in terms of the number of centres involved to return a generalisable result. However, this study is an important precursor to a larger, substantive trial and provides invaluable information that will help to ensure its success, which is ultimately

needed to provide a definitive answer to this important question.

## Author affiliations

[1]Institute of Metabolism and System Research, University of Birmingham, Birmingham, UK
[2]School of Health and Population Sciences, University of Birmingham, Birmingham, UK
[3]Birmingham Clinical Trials Unit, University of Birmingham, Birmingham, UK
[4]Nottingham Clinical Trials Unit, University of Nottingham, Nottingham, UK
[5]Obstetrics and Gynaecology, Birmingham Women's and Children's NHS Foundation Trust, Birmingham, UK

**Acknowledgements** The authors would like to thank the collaborators, women who recruited into this study, the healthcare professionals at the recruitment sites and the funders. Miss Ranee Thakar, Croydon, Mrs Preeti Jain, Walsall, Mr Christian Phillips, Basingstoke, Mr Jason Cooper, North Staffordshire, Mr Athele Khunda, South Tees, Professor Jane Daniels and Lisa Leighton. Patient and Public involvement: Rev. Ann Simons and Dr Mary Evans. Trial Oversight committee: Professor Shakila Thangaratinam, Valerie Morton, Tom Maishman and Deepali Sinha.

**Contributors** LM, JD and PML conceptualised the study as co-applicants. LM, JD and PML contributed to the intervention development and design. PML oversaw the running of the trial and all the authors contributed to the ongoing management of the trial. The LOTUS trial collaborative group collected data for the trial. LM and VC developed the data analysis plan and summarised the results. TSV and LL evaluated the qualitative data. The manuscript was drafted by TSV with contributions from PML, LM, JD, VC and LL. All the authors contributed to the interpretation of the output and revised and reviewed the paper; they are the guarantors. All authors contributed to the writing of the final manuscript and provided critical comments during revisions. All authors approved the final version prior to submission. The Birmingham Clinical Trials Unit undertook the randomisation and data management and monitoring. The authors had full access to all the data from the study. The authors vouch for the accuracy and completeness of the data and analyses. All authors read and approved the final manuscript.

**Funding** The study was funded by a grant from the National Institute for Health Research for Patient Benefit (grant number: PB-PG-0213–30126).

**Competing interests** PML received sponsorship from Astellas, Contura and Pfizer to attend meetings.

**Patient consent for publication** Not required.

**Ethics approval** This study received a favourable ethical opinion from NHS Research Ethics Committee (NRES Committee West Midlands, REC number 15/WM/0092).

**Provenance and peer review** Not commissioned; externally peer reviewed.

**Data availability statement** Data are available on reasonable request. The dataset used in this manuscript is available on request from Birmingham Clinical Trials Unit.

## ORCID iDs

Tina Sara Verghese http://orcid.org/0000-0003-3880-6416
Versha Cheed http://orcid.org/0000-0002-6713-0913

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
