## [Reviewer comments · BMJ Open]

ARTICLE DETAILS

TITLE (PROVISIONAL)	A Randomised Controlled Trial to investigate the effectiveness of Local Oestrogen Treatment in Postmenopausal Women Undergoing Pelvic Organ Prolapse Surgery (LOTUS) – a pilot study to assess feasibility of a large multi-centre trial
AUTHORS	Vergheze, Tina; Middleton, Lee; Cheed, Versha; Leighton, Lisa; Daniels, Jane; Latthe, Pallavi

VERSION 1 – REVIEW

REVIEWER	Adi Y. Weintraub Department of Obstetrics and Gynecology Soroka University Medical Center Faculty of Health Sciences Ben-Gurion University of the Negev Beer-Sheva, Israel
REVIEW RETURNED	15-Dec-2018

GENERAL COMMENTS	Thank you for the opportunity to review this manuscript. This study was designed to assess the feasibility of a multicenter randomized controlled trial comparing local estrogen treatment with no estrogen supplementation in women undergoing pelvic organ prolapse (POP) surgery. Women were randomized (1:1) to pre and post-operative estrogen or no treatment. Estrogen treatment commenced 6 weeks prior to surgery and twice weekly for 26 weeks from 6 weeks post-surgery. The main outcomes were assessment of eligibility and recruitment rates along with compliance and data completion. Prolapse specific quality of life measures included the PFDI-20 and the PGII, completed at 6 and 12 months. Clinical outcomes measured by POP Q, at 6 months post-surgery and post-operative complications. Of 325 women seeking POP surgery, 157 (48%) were found eligible. Of these, 100 (64%) were randomized, 50 to estrogen and 50 to no estrogen treatment, with 89 (44/45 respectively) ultimately having surgery. Of these, 89% returned complete questionnaires at six months and 78% reported good compliance with estrogen. No serious adverse events were attributable to estrogen use. The authors concluded that a large multicenter RCT of estrogen versus no treatment is feasible as it is possible to randomize and follow up participants with high fidelity. Four predefined feasibility criteria were met. Compliance with treatment regimens was not a barrier. A larger trial is required to definitively address the role of peri-operative estrogen supplementation. This is an interesting topic although the feasibility study is much less attractive. The manuscript is well written and could be published in its current form. I would like to raise some questions regarding the study.
--

	1. Why was the estrogen treatment started 6 weeks following surgery and not earlier? How do the authors think this interval can affect the outcomes? 2. After randomization, did the patients have an option to choose other agents (cream) or only the vagifem vaginal pessaries? 3. There seems to be a typo error on P13L48, double "the". Adi Y. Weintraub Department of Obstetrics and Gynecology Soroka University Medical Center Faculty of Health Sciences Ben-Gurion University of the Negev Beer-Sheva, Israel
--	---

REVIEWER	Sharif Ismail Brighton and Sussex University Hospitals NHS Trust
REVIEW RETURNED	28-Dec-2018

GENERAL COMMENTS	This manuscript reports a well carried out randomised controlled trial that tries to address gaps in medical evidence, with relevance to day to day clinical practice. However, it presents the findings or a pilot feasibility study, which means that the evidence provided in it is not the level to be accorded to a full randomised controlled study. The lack of a proper power calculation, based on a specified primary outcome measure, which could have been a post hoc one, makes it difficult to judge whether the lack of statistically significant difference was due to lack of power or actual lack of statistically significant difference. These points do not preclude publishing the study, as long as they are considered in the manuscript, and certainly indicating whether the study would be continued on the hope of reaching a power calculation or a full study is to follow this one. The range has no place in statistics, even when reporting published figures about the number of surgical procedures carried out for pelvic organ prolapse. Reliance on published articles in this respect would have been helpful. The centres which took part in the study should have been named in the method, rather than in the results. Providing background information about the units where the study was carried out, in terms of geographical location, catchment area, size, work load and service level, would have been helpful. The non mention of a statistical software package for statistical analysis and lack of any description of statistical tests are noted. The pelvic organ prolapse-quantification (POP-Q) stage, rather than maximum stage of pelvic organ prolapse, should have been provided in the results. The results should have started by indicating the number of cases screened for inclusion in this study and the number excluded by indication, with percentage. The findings should have been outlined in tables with text used to provide information that cannot be included in tables and/or highlight notable features. The lack of information about the outcome of the focus group interviews is noted. The mean \pm standard deviation, median [inter-quartile range] and frequency (%) are the universal formats in which these statistical parameters are presented. Tables should have had proper titles and specified the statistical tests used for each comparison. There was no need to shading in the tables.
---

	The small number of references relied upon is noted. All abbreviations should have been provided in full on first mention and this applies to the title, abstract, article summary, main text and each table / figure independently as they will be read independently. The third article (the authors, the study), rather than the first author (we/our), the passive voice and past tense are seldom improved upon in scientific manuscripts. Numbered lists are better presented as lists, rather than within paragraphs. Paragraphs should have a proper structure to have a consistent size (length). A space should have been left between sentences. Double spacing is the norm in all scientific manuscripts, and this applies to tables.
--	---

REVIEWER	Brandon Grossardt, MS (Senior Statistician) Mayo Clinic Division of Biomedical Statistics and Informatics Department of Health Sciences Research 200 First Street SW Rochester, MN, USA 55905
REVIEW RETURNED	13-Feb-2019

GENERAL COMMENTS	SUMMARY: The authors report the results of a pilot study investigating the use of low-dose estrogen post-POP repair surgery. This purpose of this report is two-fold: 1) to assess the follow-up and recruitment feasibility of such a study (with plans to eventually expand to a larger population); and 2) to report on the baseline descriptive information and the frequency of outcomes of the pilot study groups for use in planning and powering the subsequent RCT. COMMENTS: The study intent, processes, and outcomes are well described in the paper. It is well written and the methodological approaches are clear, concise, well-justified, and seem appropriate. I had no substantial concerns with regard to design, statistical methods, or other issues.
---

REVIEWER	Kevin Cooper Nhs Grampian UK
REVIEW RETURNED	12-Aug-2019

GENERAL COMMENTS	I have 2 major concerns First, it is stated that an rct is feasible, which is technically correct for the proportion willing to be randomised, but not correct for sample size required to demonstrate a meaningful difference over a meaningful length of time, as results for pop repair should be long term unless an expected short term difference in outcome is expected which given the results presented is unlikely. The sample size required may be too large to make a definitive trial expedient or cost effective. Secondly, why was there a delay to 6 weeks post open to recommence vagifem, this has missed 50% of the healing time of the repair, so is actually comparing no oestrogen with oestrogen for the last half, 6 to12 week's of the actual surgical healing process. This needs discussed
---

VERSION 1 – AUTHOR RESPONSE

Reviewers Comments	Response
Reviewer: 1 This is an interesting topic although the feasibility study is much less attractive. The manuscript is well written and could be published in its current form. I would like to raise some questions regarding the study. 1. Why was the oestrogen treatment started 6 weeks following surgery and not earlier? How do the authors think this interval can affect the outcomes? 2. After randomization, did the patients have an option to choose other agents (cream) or only the vagifem vaginal pessaries?	Thank you for your comments and suggestions. 1. Post operative oestrogen therapy was commenced after 6 weeks of surgery, as women were reluctant to insert intra vaginal oestrogen pessaries during this period. Patient group discussion expressed that they felt they would be confident to insert an intra vaginal pessary 6 weeks after surgery and not earlier as they were worried about disturbing the healing process. 2. No, the patient did not have an option. A survey and a focus group interview was conducted by the research team prior to commencing the feasibility study among women with pelvic organ prolapse. The women voiced that they would prefer vaginal oestrogen pessaries rather than oestrogen cream. Therefore, all the patients randomised to treatment arm were given vagifem vaginal pessaries.

3. There seems to be a typo error on P13L48, double "the".	3. Thank you for pointing this out, we apologise for the typo. Please find rectification in the manuscript.
Reviewer: 2 This manuscript reports a well carried out randomised controlled trial that tries to address gaps in medical evidence, with relevance to day to day clinical practice. However, it presents the findings for a pilot feasibility study, which means that the evidence provided in it is not the level to be accorded to a full randomised controlled study. The lack of a proper power calculation, based on a specified primary outcome measure, which could have been a post hoc one, makes it difficult to judge whether the lack of statistically significant difference was due to lack of power or actual lack of statistically significant difference. These points do not preclude publishing the study, as long as they are considered in the manuscript, and certainly indicating whether the study would be continued on the hope of reaching a power calculation or a full study is to follow this one. The range has no place in statistics, even when reporting published figures about the number of surgical procedures carried out for pelvic organ prolapse. Reliance on published articles in this respect would have been helpful. The centres which took part in the study should have been named in the method, rather than in the results. Providing background information about the units where the study was carried out, in terms of geographical location, catchment area, size, work load and service level, would have been helpful.	Thank you for your comment. As stated in the manuscript, this feasibility study is first step before conducting a adequately powered RCT. It did answer the primary question in the context of whether it is possible to do a large scale multicenter RCT. The team is currently focusing on obtaining funding for a proposed definitive RCT based on the results of the feasibility study.

The non mention of a statistical software package for statistical analysis and lack of any description of statistical tests are noted.

We have now described within the manuscript (page 7 and page 11) that a combination of district general hospital and university teaching hospital urogynaecology units took part in the study. We do not think it is necessary to describe each unit's name, location in detail in the manuscript but have mentioned all the participating centres' names at the end.

As described in the statistical analysis section we did not undertake statistical tests due to the nature of the objectives of the study. This is considered appropriate for a pilot study (Ref: Kannan et al. Pilot studies: Are they appropriately reported? *Perspect Clin Res.* 2015 Oct-Dec; 6(4): 207–210.)

We have added the name of the statistical software used to generate estimates in the statistical analysis section. Amendment now seen on page 9.

The pelvic organ prolapse-quantification (POP-Q) stage, rather than maximum stage of pelvic organ prolapse, should have been provided in the results.

The results should have started by indicating the number of cases screened for inclusion in

this study and the number excluded by indication, with percentage. The findings should have been outlined in tables with text used to provide information that cannot be included in tables and/or highlight notable features.

The lack of information about the outcome of the focus group interviews is noted.

The mean \pm standard deviation, median [inter-quartile range] and frequency (%) are the universal formats in which these statistical parameters are presented. Tables should have had proper titles and specified the statistical tests used for each comparison. There was no need to shading in the tables.

The small number of references relied upon is noted.

All abbreviations should have been provided in full on first mention and this applies to the title, abstract, article summary, main text and each table / figure independently as they will be read independently. The third article (the authors, the study), rather than the first author (we/our), the passive voice and past tense are seldom improved upon in scientific manuscripts. Numbered lists are better presented as lists, rather than within paragraphs. Paragraphs should have a proper structure to have a consistent size (length). A space should have been left between sentences. Double spacing is the norm in all scientific manuscripts, and this applies to tables.

The authors of PROSPECT and VUE study utilized the POP-Q study to report maximum stage of prolapse.

We have mentioned these results on page 11.

The qualitative arm of this study is being peer reviewed in another journal

We are happy to comply with any editorial requests for formatting tables. As per previous response, no tests were undertaken.

All abbreviations have now been rechecked and provided in full in first mention. The manuscript has now been re formatted and spacing as per the journals specifications.

Reviewer: 3	
The study intent, processes, and outcomes are well described in the paper. It is well written and the methodological approaches are clear, concise, well-justified, and seem appropriate. I had no substantial concerns with regard to design, statistical methods, or other issues.	Thank you for your comments.

VERSION 2 – REVIEW

REVIEWER	Dr. Sharif Ismail Brighton and Sussex University Hospitals NHS Trust United Kingdom
REVIEW RETURNED	16-Nov-2019

GENERAL COMMENTS	This is the second submission of this manuscript, which continues to suffer from limitations that preclude its publication. The abstract should have been started on a new page. “Primary and secondary outcome measures” are the proper term that should have been used. “Clinical outcome measures were POP, as assessed by POP quantification system 6 months after surgery and post-operative complications” would have been better than what is written under the subheading “outcome measures”, in the abstract section. “to judge” should have been deleted from the first bullet point under the subheading “Strengths and limitations of this study”, in the article summary section. The introduction should have started by defining pelvic organ prolapse, according to the latest standardisation and terminology document of the International Urogynecological Association (IUGA). “data” is plural and should have attracted the plural verb “are” rather than singular one “is”, in the last sentence of the first paragraph of the introduction. “Moreover, if this use reduces in failure and recurrence rates, this would save cost.” Would have been better than what is written in the last sentence of the fourth paragraph of the introduction. The fifth paragraph of the introduction does not include any supporting references. “The aim of this feasibility pilot trial was” would have been better than what is written at the start of the sixth paragraph of the introduction. Methodological detail belongs to the method, rather than the introduction. A reference should have been provided for the CONSORT check list for reporting a pilot trial, referred to in the first paragraph of the method. A reference should have been provided for the qualitative study evaluating women’s perspective, which was presented at “the” Annual Scientific Meeting of the International Continence Society in 2017. Descriptive statistics should have been described. The results should have started by indicating the number of cases screened for inclusion in this study as well as the number excluded by indication / dropped out at various stages, with percentage.
---

This would have been best provided in a CONSORT flow chart. The findings should have been outlined in tables with text used to provide information that cannot be included in tables and/or highlight notable features. The average has no meaning in statistics. Appearances of similarity, or difference, are not terms to be seen in scientific literature and certainly in reports of randomised controlled trials. The mean \pm standard deviation, median [inter-quartile range] and frequency (%) are the universal formats in which these statistical parameters are presented. A comparison of base line features between the 2 groups should have been carried out, with the statistical tests used for comparison being included in tables 1 and 2. Continuous variables should be checked for normality before deciding whether to use parametric or non-parametric tests for their description and comparison. There was no need to shading in the tables. Questionnaire scores and pelvic organ prolapse-quantification values should have been compared using paired comparative statistical tests.

“Majority” is not the precise term to be expected in scientific literature and certainly in reports of randomised controlled trials. The reflection on the study findings in the light of available literature should have focused on the study findings. The statement that “the overall quality of evidence was poor”, in the second paragraph of the discussion, should have qualified and elaborated upon. Probably, this information should have been better outlined in the introduction. “the current study (LOTUS)” would have been better than “the LOTUS study” in the third paragraph of the discussion. “both” would have been better than “the” in the sixth sentence of the third paragraph of the discussion. The statement made in the first sentence of the fourth paragraph of the discussion required supporting references.

The average number of references for an original paper is 25.

It is better not to start sentences with numbers in digits. All abbreviations should have been provided in full on first mention and this applies to the title, key words, abstract, article summary, main text and each table/figure independently as they would be read independently. Paragraphs should have had proper structure to have appropriate consistent length (size). The third article (the authors/study), rather than the first article (we/our), passive voice and past tense are seldom improved upon in scientific manuscripts. “hospitals” would have been better than “hospital”, in the first sentence of the third paragraph of the method. Extra lines between paragraphs should have been deleted. “Focus group interviews” would have been better than “Focus groups interview”, in the (one sentence) final paragraph of the method. Paragraphs should have had proper structure to have appropriate consistent length (size). The use of “the” and punctuation need attention. The reference style within text should have been consistent, in terms of having a space before the opening bracket. “cost” would have been better than “costs”, in the fourth sentence of the fourth paragraph of the discussion. “the” would have been better than “their”, “showed” would have been better than “justified”, “could” would have been better than “will” and “achieved” would have been better than “achievable”, in the second sentence of the first paragraph under the subheading “Strengths”. “is ultimately” would have been better than “ultimately is”. A full stop was missing after “design”, in the second sentence under the heading “Authors’

	contributions”, where repetitions need to be eliminated. “blood” should have been removed from before “thrombo-embolism”, in the penultimate comparison item in table 2.
--	--

REVIEWER	kevin cooper Dept of obstetrics and gynaecology Aberdeen Royal Infirmary UK
REVIEW RETURNED	02-Nov-2019

GENERAL COMMENTS	The reviewers queries have been answered and there is some disagreement, which is not unhealthy. I'm still not fully convinced at using 6 weeks post surgery as the time for recommencing E2 treatment on the basis of patient opinion. Surely if undertaking what would need to be a very large rct (as no meaningful difference detected in this feasibility study) with FU beyond 2 years for meaningful outcomes, that the actual effects of the intervention are measured. Patients would not come to physical harm by restarting the oestrogen 2 weeks post op, or even 4, which may have an important effect on the early stages of healing post surgery. I think this should at least be addressed in discussion along with what is the most meaningful primary outcome, objective return of same compartment, re-operation / pessary for same compartment, patient reported symptom outcome etc. The proposed sample size required and timescale to complete based on number of centres. This would then be an informative paper which would generate interest and discussion
--

VERSION 2 – AUTHOR RESPONSE

Reviewer: 2

Reviewer Name: Dr. Sharif Ismail

Institution and Country: Brighton and Sussex University Hospitals NHS Trust
United Kingdom

Please state any competing interests or state 'None declared': None

Please leave your comments for the authors below

This is the second submission of this manuscript, which continues to suffer from limitations that preclude its publication.

The abstract should have been started on a new page.

“Primary and secondary outcome measures” are the proper term that should have been used.
“Clinical outcome measures were POP, as assessed by POP quantification system 6 months after surgery and post-operative complications” would have been better than what is written under the subheading “outcome measures”, in the abstract section.

“to judge” should have been deleted from the first bullet point under the subheading “Strengths and limitations of this study”, in the article summary section.

The introduction should have started by defining pelvic organ prolapse, according to the latest

standardisation and terminology document of the International Urogynecological Association (IUGA). “data” is plural and should have attracted the plural verb “are” rather than singular one “is”, in the last sentence of the first paragraph of the introduction. “Moreover, if this use reduces in failure and recurrence rates, this would save cost.” Would have been better than what is written in the last sentence of the fourth paragraph of the introduction. The fifth paragraph of the introduction does not include any supporting references. “The aim of this feasibility pilot trial was” would have been better than what is written at the start of the sixth paragraph of the introduction. Methodological detail belongs to the method, rather than the introduction. A reference should have been provided for the CONSORT check list for reporting a pilot trial, referred to in the first paragraph of the method. A reference should have been provided for the qualitative study evaluating women’s perspective, which was presented at “the” Annual Scientific Meeting of the International Continence Society in 2017. Descriptive statistics should have been described.

The results should have started by indicating the number of cases screened for inclusion in this study as well as the number excluded by indication / dropped out at various stages, with percentage. This would have been best provided in a CONSORT flow chart. The findings should have been outlined in tables with text used to provide information that cannot be included in tables and/or highlight notable features. The average has no meaning in statistics. Appearances of similarity, or difference, are not terms to be seen in scientific literature and certainly in reports of randomised controlled trials. The mean \pm standard deviation, median [inter-quartile range] and frequency (%) are the universal formats in which these statistical parameters are presented. A comparison of base line features between the 2 groups should have been carried out, with the statistical tests used for comparison being included in tables 1 and 2. Continuous variables should have been checked for normality before deciding whether to use parametric or non-parametric tests for their description and comparison. There was no need to shading in the tables. Questionnaire scores and pelvic organ prolapse-quantification values should have been compared using paired comparative statistical tests.

“Majority” is not the precise term to be expected in scientific literature and certainly in reports of randomised controlled trials. The reflection on the study findings in the light of available literature should have focused on the study findings. The statement that “the overall quality of evidence was poor”, in the second paragraph of the discussion, should have qualified and elaborated upon. Probably, this information should have been better outlined in the introduction. “the current study (LOTUS)” would have been better than “the LOTUS study” in the third paragraph of the discussion. “both” would have been better than “the” in the sixth sentence of the third paragraph of the discussion. The statement made in the first sentence of the fourth paragraph of the discussion required supporting references.

The average number of references for an original paper is 25.

It is better not to start sentences with numbers in digits. All abbreviations should have been provided in full on first mention and this applies to the title, key words, abstract, article summary, main text and each table/figure independently as they would be read independently. Paragraphs should have had proper structure to have appropriate consistent length (size). The third article (the authors/study), rather than the first article (we/our), passive voice and past tense are seldom improved upon in scientific manuscripts. “hospitals” would have been better than “hospital”, in the first sentence of the third paragraph of the method. Extra lines between paragraphs should have been deleted. “Focus group interviews” would have been better than “Focus groups interview”, in the (one sentence) final paragraph of the method. Paragraphs should have had proper structure to have appropriate consistent length (size). The use of “the” and punctuation need attention. The reference style within text should have been consistent, in terms of having a space before the opening bracket. “cost” would have been better than “costs”, in the fourth sentence of the fourth paragraph of the discussion. “the” would have been better than “their”, “showed” would have been better than “justified”, “could” would

have been better than “will” and “achieved” would have been better than “achievable”, in the second sentence of the first paragraph under the subheading “Strengths”. “is ultimately” would have been better than “ultimately is”. A full stop was missing after “design”, in the second sentence under the heading “Authors’ contributions”, where repetitions need to be eliminated. “blood” should have been removed from before “thrombo-embolism”, in the penultimate comparison item in table 2.

Reviewer: 4

Reviewer Name: kevin cooper

Institution and Country: Dept of obstetrics and gynaecology
 Abereen Royal Infirmary
 UK

Please state any competing interests or state ‘None declared’: none

Please leave your comments for the authors below

The reviewers queries have been answered and there is some disagreement, which is not unhealthy. I'm still not fully convinced at using 6 weeks post surgery as the time for recommencing E2 treatment on the basis of patient opinion. Surely if undertaking what would need to be a very large rct (as no meaningful difference detected in this feasibility study) with FU beyond 2 years for meaningful outcomes, that the actual effects of the intervention are measured. Patients would not come to physical harm by restarting the oestrogen 2 weeks post op, or even 4, which may have an important effect on the early stages of healing post surgery.

I think this should at least be addressed in discussion along with what is the most meaningful primary outcome, objective return of same compartment, re-operation / pessary for same compartment, patient reported symptom outcome etc.

The proposed sample size required and timescale to complete based on number of centres.

This would then be an informative paper which would generate interest and discussion

VERSION 3 – REVIEW

REVIEWER	Dr. Sharif Ismail Brighton and Sussex University Hospitals NHS Trust United Kingdom
REVIEW RETURNED	25-Feb-2020

GENERAL COMMENTS	This is an improved manuscript. However, there are few minor points that will improve the publication. The authors are congratulated on their excellent work and wished well with the full randomised controlled trial. Explaining that an external pilot means that the data are not added to the proper trial would be helpful to those who are not aware of this term. The second sentence could have been split into two, with the second one starting with “However, the diagnosis of ...”. “the” should have been deleted from before “women”, in the penultimate sentence of the third paragraph of the introduction. “such randomised controlled trials” would have been better than “research”, in the second sentence of the last paragraph of the introduction.
--

	The findings of the Cochrane review, which is mentioned in a one sentence paragraph, should have been outlined more clearly. The use before surgery is distinct from use for prevention and treatment of pelvic organ prolapse. The fact that reduction in urinary tract infection in the first 2 weeks after surgery was suggested, though not unequivocally proven, is worth flagging up, as a potential benefit from using oestrogen before surgery for pelvic organ prolapse. The following randomised controlled trial should have been covered in the introduction and relied upon in the discussion. Karp, Deborah R. MD; Jean-Michel, Marjorie MD; Johnston, Yasmin MD; Suciu, Gabriel PhD; Aguilar, Vivian C. MD; Davila, G. Willy MD (2012) A Randomized Clinical Trial of the Impact of Local Estrogen on Postoperative Tissue Quality After Vaginal Reconstructive Surgery, Female Pelvic Medicine & Reconstructive Surgery, Volume 18 - Issue 4 - p 211-215 doi: 10.1097/SPV.0b013e31825e6401 The address and/or the website address of the manufacturer of the statistical software used for statistical analysis should have been provided. The findings should have been outlined in tables with text used to provide information that cannot be included in tables and/or highlight notable features. The mean \pm standard deviation, median [inter-quartile range], frequency (%) [95% confidence interval with – between figures] are the universal formats in which these statistical parameters are presented. The tables should have included the comparison between the groups (test name and P value). It would have been better to put the significant differences in bold, for the readers to notice them. A comma is missing after study, in the first sentence of the first paragraph of the discussion. “efficiently” is better deleted from the second sentence of the strengths section of the discussion. A “,” is better added before “without”, in the third sentence of the limitations section of the discussion. “post randomisation” would have been better after “surgery”, in the first sentence of the second paragraph of the limitations section of the discussion. It is better not to start sentences with numbers in digits. Paragraphs should have had proper structure to have appropriate consistent length (size). The third article (the authors/study), rather than the first article (we/our), passive voice and past tense are seldom improved upon in scientific manuscripts. A comma is best added after feasible, in the first sentence of the conclusion, of the abstract. Having a full stop, or none, at the end of each bullet point, under the strengths and limitations is consistent. Consistent leaving, or not leaving, a space before the reference number opening bracket is better.
--	--

VERSION 3 – AUTHOR RESPONSE

Reviewers comments	Response
Reviewer: 2	

Reviewer Name: Dr. Sharif Ismail

Institution and Country:

Brighton and Sussex University Hospitals NHS Trust

United Kingdom

Please state any competing interests or state 'None declared': None declared

Please leave your comments for the authors below

This is an improved manuscript. However, there are few minor points that will improve the publication. The authors are congratulated on their excellent work and wished well with the full randomised controlled trial.

Explaining that an external pilot means that the data are not added to the proper trial would be helpful to those who are not aware of this term.

The second sentence could have been split into two, with the second one starting with "However, the diagnosis of ...". "the" should have been deleted from before "women", in the penultimate sentence of the third paragraph of the introduction. "such randomised controlled trials" would have been better than "research", in the second sentence of the last paragraph of the introduction.

The findings of the Cochrane review, which is mentioned in a one sentence paragraph, should have been outlined more clearly. The use before surgery is distinct from use for prevention and treatment of pelvic organ prolapse. The fact that reduction in urinary tract infection in the first 2 weeks after surgery was suggested, though not unequivocally proven, is worth flagging up, as a potential benefit from using oestrogen before surgery for pelvic organ prolapse.

We thank the reviewer for his comments and we have further improved the manuscript as per his suggestions.

We have defined external pilot in the methods section on page 7.

Thank you for your suggestions. We have amended these sentences.

This has been elaborated and highlighted in the discussion section

The following randomised controlled trial should have been covered in the introduction and relied upon in the discussion.

Karp, Deborah R. MD; Jean-Michel, Marjorie MD; Johnston, Yasmin MD; Suci, Gabriel PhD; Aguilar, Vivian C. MD; Davila, G. Willy MD (2012) A Randomized Clinical Trial of the Impact of Local Estrogen on Postoperative Tissue Quality After Vaginal Reconstructive Surgery, Female Pelvic Medicine & Reconstructive Surgery, Volume 18 - Issue 4 - p 211-215 doi: 10.1097/SPV.0b013e31825e6401

The address and/or the website address of the manufacturer of the statistical software used for statistical analysis should have been provided.

The findings should have been outlined in tables with text used to provide information that cannot be included in tables and/or highlight notable features. The mean \pm standard deviation, median [interquartile range], frequency (%) [95% confidence interval with – between figures] are the universal formats in which these statistical parameters are presented. The tables should have included the comparison between the groups (test name and P value). It would have been better to put the significant differences in bold, for the readers to notice them.

A comma is missing after study, in the first sentence of the first paragraph of the discussion.

“efficiently” is better deleted from the second sentence of the strengths section of the discussion.

A “,” is better added before “without”, in the third sentence of the limitations section of the discussion.

Reference added and relied on in the discussion, thank you for the suggestion

The statistical software is stated in the manuscript at the end of statistical analysis section first paragraph

Statistical testing of variables is not appropriate for outcomes in pilot studies as the study has not been powered to reject any hypothesis. We have focussed on reporting estimated differences and 95% confidence intervals as recommended as good practice for reporting pilot studies (ref: Design and analysis of pilot studies: recommendations for good practice. Lancaster, Gillian A; Dodd, Susanna; Williamson, Paula R. Journal of evaluation in clinical practice. , 2004, Vol.10(2), p.307-312).

“post randomisation” would have been better after “surgery”, in the first sentence of the second paragraph of the limitations section of the discussion.

It is better not to start sentences with numbers in digits.

Paragraphs should have had proper structure to have appropriate consistent length (size). The third article (the authors/study), rather than the first article (we/our), passive voice and past tense are seldom improved upon in scientific manuscripts.

A comma is best added after feasible, in the first sentence of the conclusion, of the abstract.

Having a full stop, or none, at the end of each bullet point, under the strengths and limitations is consistent. Consistent leaving, or not leaving, a space before the reference number opening bracket is better.

A comma has been added after study, in the first sentence of the first paragraph of the discussion

“efficiently” has been deleted from the second sentence of the strengths section of the discussion.

“,” has been added before without in the third sentence of the limitations section of the discussion.

Thank you for the suggestion.

We have rechecked the manuscript and made sure that no sentence begins with numbers in digits.

Paragraphs have been structures to appropriate and consistent length. The third article has been used appropriately.

A comma has now been added. Full stop placed at the end of each bullet point under strengths and limitation.

We have left no space before the reference number opening bracket in the manuscript.

VERSION 4 – REVIEW

REVIEWER	Dr. Sharif Ismail Brighton and Sussex University Hospitals NHS Trust United Kingdom
REVIEW RETURNED	02-Jun-2020

GENERAL COMMENTS	This is the third submission of this manuscript which continues to suffer from limitations noted before. These were highlighted in the best effort to assist the authors in getting the manuscript ready for publication. However, some of the comments provided earlier continue to be ignored. It is for the Editor to decide if these are worth correcting or the paper are acceptable as it is. The findings of the studies that looked at the value of oestrogen for pelvic organ prolapse, especially in relation to its local (vaginal) use alongside prolapse surgery should have been outlined in more detail in the introduction. This would have shown the nature of the likely benefit to advocate the use of local oestrogen alongside surgery for pelvic organ prolapse. It would have been good to include the website of the manufacturer of the statistical software used for statistical analysis of data. The results should have started by indicating the number of cases screened for inclusion in this study and the number excluded by indication, with percentage. The findings should have been outlined in tables with text used to provide information that cannot be included in tables and/or highlight notable features. This would have enabled more detailed outline of the findings of the studies that looked at the value of local oestrogen with surgery for pelvic organ prolapse. The average has no place in statistics. Similarity is different from lack of statistically significant difference and requires special power calculation, which was not applied in this study. The mean \pm standard deviation, median [inter-quartile range] and frequency (%) are the universal formats in which these statistical parameters are presented. All tables should have included a comparison between the two arms for each variable, with the test used in the comparison. This would have enabled appreciating any difference at baseline features as well as at follow up points. Table 5 shows the pelvic organ prolapse-quantification (POP-Q) as a continuous variable. It would have been good to look at it also as a categorical one. This would have enabled comparing stages of pelvic organ prolapse. The addition of one randomised controlled trial that looked at local (vaginal) oestrogen use alongside surgery for pelvic organ prolapse meant that there are now four, rather than three, such studies mentioned in the manuscript. All abbreviations should have been provided in full on first mention and this applies to different parts of the manuscript individually as they would be read independently. RCT is mentioned for the first time in the second paragraph of the introduction, without being provided in full earlier in the main text of the manuscript. Whilst this abbreviation is common knowledge nowadays, it might change in years to come, when a publication will still be available and should be understood at any point in the future. Paragraphs should have had a proper structure to have a consistent length. For example, the second paragraph of the introduction is formed of one sentence and follows a rather long paragraph.
--

	The third article (the authors/study), rather than the first article (we/our), passive voice and past tense are seldom improved upon in scientific manuscripts. “we” and “our” can be seen throughout the manuscript. All sentences should have been ended with a full stop.
--	--